# Moral Distress Scores of Nurses Working in Intensive Care Units for Adults Using Corley’s Scale: A Systematic Review

**DOI:** 10.3390/ijerph191710640

**Published:** 2022-08-26

**Authors:** Noemi Giannetta, Giulia Villa, Loris Bonetti, Sara Dionisi, Andrea Pozza, Stefano Rolandi, Debora Rosa, Duilio Fiorenzo Manara

**Affiliations:** 1School of Nursing, UniCamillus—Saint Camillus International University of Health and Medical Sciences, 00131 Rome, Italy; 2Center for Nursing Research and Innovation, Vita-Salute San Raffaele University, 20132 Milan, Italy; 3Department of Nursing, Nursing Research Centre, Ente Ospedaliero Cantonale (EOC), 6500 Bellinzona, Switzerland; 4Department of Business Economics, Health and Social Care, University of Applied Sciences and Arts of Southern Switzerland, 6928 Manno, Switzerland; 5Department of Biomedicine and Prevention, University of Rome “Tor Vergata”, 00133 Rome, Italy; 6Department of Medical Sciences, Surgery and Neurosciences, University of Siena, 53100 Siena, Italy; 7IRCCS San Raffaele Scientific Institute, 20132 Milan, Italy; 8Department of Cardiovascular, Neural, and Metabolic Sciences, IRCCS Istituto Auxologico Italiano, 20149 Milan, Italy

**Keywords:** ethical conflict, health professionals, moral courage, moral distress, nurses, nursing, systematic review

## Abstract

Background: No systematic review in the literature has analyzed the intensity and frequency of moral distress among ICU nurses. No study seems to have mapped the leading personal and professional characteristics associated with high levels of moral distress. This systematic review aimed to describe the intensity and frequency of moral distress experienced by nurses in ICUs, as assessed by Corley’s instruments on moral distress (the Moral Distress Scale and the Moral Distress Scale–Revised). Additionally, this systematic review aimed to summarize the correlates of moral distress. Methods: A systematic search and review were performed using the following databases: Cumulative Index to Nursing and Allied Health Literature (CINAHL), the National Library of Medicine (MEDLINE/PubMed), and Psychological Abstracts Information Services (PsycINFO). The review methodology followed PRISMA guidelines. The quality assessment of the included studies was conducted using the Newcastle–Ottawa Scale. Results: Findings showed a moderate level of moral distress among nurses working in ICUs. The findings of this systematic review confirm that there are a lot of triggers of moral distress related to patient-level factors, unit/team-level factors, or system-level causes. Beyond the triggers of moral distress, this systematic review showed some correlates of moral distress: those nurses working in ICUs with less work experience and those who are younger, female, and intend to leave their jobs have higher levels of moral distress. This systematic review’s findings show a positive correlation between professional autonomy, empowerment, and moral distress scores. Additionally, nurses who feel supported by head nurses report lower moral distress scores. Conclusions: This review could help better identify which professionals are at a higher risk of experiencing moral distress, allowing the early detection of those at risk of moral distress, and giving the organization some tools to implement preventive strategies.

## 1. Introduction

The first author to define moral distress was Andrew Jameton (1984). In his 1984 nursing ethics textbook, Jameton (1984) classified three experiences related to ethical problems that a nurse can meet in a clinical setting: moral uncertainty, moral dilemmas, and moral distress. Focusing on the last, moral distress arises when “one knows the right thing to do, but institutional constraints make it nearly impossible to pursue the right course of action” [1]. Based on this definition, the concept of moral distress was confined to the nursing population; however, several studies suggest that it includes all healthcare professionals in all healthcare settings. Therefore, discussions about broadening this concept have been based on the observation that moral distress can be experienced as a reply to any moral conflict [2].

In recent years, literature on moral distress has steadily increased. The most is known about its triggers in healthcare systems and its consequences on healthcare professionals’ physical or psychological well-being. A recent integrative review classified triggers of moral distress into three groups: (a) patient-level factors, including the patient and/or family (e.g., the family requires aggressive therapy or performing treatments that induce excessive pain); (b) the unit/team level issues that occur when there is poor communication in the unit or inadequate collaboration between team members; and (c) system-level causes, including those that occur outside the unit, such as chronic poor staffing, pressure from administrators to reduce costs, a lack of adequate resources, and so on [3,4,5,6,7,8]. Therefore, morally distressing events are present in all types of wards or healthcare settings [9,10,11,12,13,14]. However, several studies have focused on the moral distress experienced by intensive care nurses. In intensive care units (ICUs), nurses make active treatment and end-of-life decisions more frequently than in other settings [15,16]. Beyond risk factors, the experience of moral distress influences nursing practice and has several consequences; some studies have linked moral distress to burnout and intention to leave [9,17,18], while others have linked moral distress to job satisfaction, a sense of coherence, professional autonomy, and ethical climate [17,18]. 

The findings of these studies are highly heterogeneous. Indeed, moral distress in each study was assessed using a different instrument. A recent literature review identified all available instruments for measuring moral distress [10]. Based on this, there are two main groups of instruments for assessing moral distress: (1) Corley’s instruments on moral distress (the Moral Distress Scale and the Moral Distress Scale–Revised [5]) and (2) instruments not directly derived from Corley’s moral distress theory (the Moral Distress Thermometer [19], the Moral Distress Risk Scale [20], the Ethical Stress Scale [21] or the Moral Distress in Dementia Care Survey [22]. The first set is the most frequently studied and used in different clinical settings and healthcare populations.

As shown by Giannetta et al. [10], Corley’s instruments on moral distress, i.e., the Moral Distress Scale [23], and the Moral Distress Scale–Revised [5], allow the intensity and frequency of moral distress to be assessed. In terms of intensity, participants rate each item on a Likert scale, measuring how distressing the individual felt that the event was when it occurred (from “not at all” to “very high”). In terms of frequency, participants rate each item on a Likert scale, measuring how often moral distress occurred (from “not at all” to “very high”). The frequency and intensity scores of moral distress are summed to create an overall moral distress score: higher scores reflect more severe levels of distress.

### Rationale and Objective

No systematic review in the literature has analyzed the intensity and frequency of moral distress among ICU nurses. Furthermore, no study seems to have mapped the main personal and professional characteristics associated with high levels of moral distress. The instruments developed by Corley and colleagues [5,23] are the most popular instruments used in the literature; they provide a clear and numerical assessment of the levels of moral distress in intensity and frequency. 

Indeed, according to Corley’s instrument, the study of moral distress can be read both from an intensity and frequency perspective. According to Corley et al. [24], there is a moderate correlation between the intensity and frequency of moral distress, and this correlation indicates that these two concepts are different. Consequently, using Corley’s instrument to measure moral distress could help identify the extent of the phenomenon and its severity. 

Moreover, no systematic review seems to have studied the relationship between the correlates of moral distress and Corley’s instrument scores in ICUs. Doing so would provide the possibility of identifying the variables associated with different levels of moral distress intensity and frequency. It would be very useful to identify those at risk early on and take preventive measures. 

Therefore, this systematic review aimed to describe the intensity and frequency of moral distress experienced by nurses in ICUs, as assessed by Corley’s instruments on moral distress (the Moral Distress Scale and the Moral Distress Scale–Revised). Furthermore, this systematic review aimed to summarize all of the factors correlated to moral distress. Based on that, this systematic review sought to answer these research questions: what are the frequency and intensity levels of moral distress among nurses working in ICUs according to Corley’s scale? Additionally, what are the factors related to moral distress in this setting (ICUs)? What are the triggers that activate moral distress among nurses working in ICUs?

## 2. Materials and Methods

### 2.1. Literature Search

This systematic review was conducted using the PRISMA (Preferred Reporting Items for Systematic Reviews and Meta-Analyses) guidelines [25]. Before performing the review, the objectives and methodology were reported in a protocol that can be requested from the corresponding author.

### 2.2. Eligibility Criteria, Information Sources, and Search Strategy

A search strategy was drawn up to identify studies focusing on moral distress scores in ICUs according to the population and their problem, exposure, outcomes, or themes (PEOS) system [26]. The population included registered nurses; the exposure was moral distress, the outcomes were the scores of moral distress among nurses in intensity and frequency, and the settings were ICUs for adults. Appendix A shows the search strategies used to retrieve studies from the following databases: the Cumulative Index to Nursing and Allied Health Literature (CINAHL), the National Library of Medicine (MEDLINE/PubMed), and Psychological Abstracts Information Services (PsycINFO). The search of the databases was conducted during the second week of November 2020. 

Table 1 shows the PEOS strategy with the inclusion and exclusion criteria. Specifically, the inclusion criteria were: Articles conducted with registered nurses working in intensive care units for adults.Articles that assessed moral distress scores using the set of Corley’s instruments on moral distress. These instruments had to be validated and adapted to the context in which the study was conducted (i.e., the article had to report the main psychometric properties of the instrument, such as the indices of reliability and factorial validity).Studies with a quantitative method design (observational, cross-sectional studies).Randomized controlled trials (RCTs), longitudinal, or case-control studies were included if they measured moral distress scores at baseline (before any intervention) using the set of Corley’s instruments on moral distress. Measures of moral distress after the intervention were not analyzed.Articles that reported moral distress mean scores, standard deviations, and sample sizes. The researchers contacted the studies’ authors to find this information if data were missing. If the authors did not reply, the article was excluded.Articles published from 2000 to November 2020. The authors decided to review work from the last 20 years because the first theorization by Corley dates to 2001 [23].Articles published in peer-review journals.Publications in English or Italian.

The exclusion criteria were:Studies with a qualitative method design.Systematic reviews, meta-analyses, or any other type of review, single-case studies, letters/opinion papers/commentaries.Articles with a validation or cross-cultural adaptation design—if they only reported the findings related to the psychometric properties, they did not report the moral distress data (means, standard deviations, and sample sizes), or the authors did not respond when contacted to provide such data.Articles focusing on healthcare professionals other than registered nurses (physicians, assistants, pharmacists, etc.) working in intensive care units.Articles focusing on registered nurses working in hospitals or primary care units (not in intensive care units).Articles focusing on moral distress scores as assessed by another instrument (the Moral Distress Thermometer [19], the Moral Distress Risk Scale [20], the Ethical Stress Scale [21], or the Moral Distress in Dementia Care Survey [22].

### 2.3. Selection and Data-Collection Process

All records retrieved from the above-mentioned databases were collected in an Excel spreadsheet. Two authors (N.G. and G.V.) independently screened the titles and abstracts for all of the records retrieved from the databases, and they assessed them using the eligibility criteria. In the case of disagreements, a third author (D.F.M.) participated in achieving consensus. 

After this screening, the authors searched the full texts of the records assessed for eligibility. All authors independently screened the full texts of these records. Discrepancies between the reviewers were resolved by discussion in research meetings with authors who were not involved in this process. In this systematic review, there were no discrepancies between the authors. However, when the screening process for a study resulted in exclusion, the reasons were documented. 

### 2.4. Data Extraction 

Data were extracted into standard data-extraction forms independently by five authors (D.R., G.V., N.G., L.B. and S.D.), including the following information: Study characteristics (study citation, study design, and country)Type of instruments administered (the name of the version of Corley’s instrument)Data-collection methods (based on web survey or paper questionnaire)Scores of moral distress (the mean score scale; the mean of frequency; the mean of intensity, and its standard deviation)Characteristics of the included participants (sample size; the ratio of female-to-male in the sample; the mean age and its standard deviation; the mean number of years of job experience as a nurse, its standard deviation, and the range) (see Table 2 and Table 3).

### 2.5. Quality Assessment 

Two authors (N.G. and G.V.) independently conducted the quality assessment of the included studies using the Newcastle–Ottawa Scale (NOS). This scale, designed for cohort studies (Wells et al., 2009), was customized for cross-sectional studies [42,43,44]. The latest version was used in this systematic review. Any disagreement was resolved by a third author (D.F.M). The total possible score is 10 for a cross-sectional study. Following Herzog et al. (2013), a study with a score of 9–10 was considered to be of “very good quality”; a study with a score of 7–8 was considered to be of “good quality”; a study with a score of 5–6 was considered to be of “satisfactory quality”; and a study with a score of less than 5 was considered to be of “unsatisfactory quality” [42]. 

## 3. Results

### 3.1. Study Selection

According to PRISMA guidelines [25], the process for study selection is shown in Appendix A. Of the 3441 records identified from the databases, 1539 were found to be duplicates and thus excluded. According to PRISMA guidelines, a further 1863 were excluded after the reading of the titles and the abstracts. In the case of doubt, the full text of the article was read, and a decision was made. The full texts of the remaining 39 articles were reviewed; 25 of these did not meet the inclusion criteria; the remaining 17 were included in this systematic review.

### 3.2. General Characteristics of the Studies 

Table 2 shows the main characteristics of the included studies. Four studies were conducted in Canada [27,28,29], four in Iran [30,31,40,41], five in the United States of America [32,33,36,37,38], two in Italy [11,14]; one in Israel [39], and one study was a multi-center study conducted in Croatia, Cyprus, the Netherlands, Slovenia, the United Kingdom, and Belgium [34]. A total of 16 of the 17 studies had a descriptive cross-sectional design, and only one was a quasi-experimental study [33].

### 3.3. General Characteristics of the Instruments and Data Collection

According to the inclusion and exclusion criteria, all included studies used the Moral Distress Scale based on Corley’s theory. Most of them used the Moral Distress Scale–Revised with 21 items [27,28,29,30,31,32,33,34,35]; 4 of them used the Moral Distress Scale–Revised with 38 items [36,37,38,39]; 2 studies used the Moral Distress Scale–Revised with 18 items [14,40]. Lamiani et al. [11] and Asayesh et al. [41] used the Moral Distress Scale–Revised with 14 and 30 items, respectively. All of the instruments used in the included studies have been validated.

Paper-and-pencil questionnaires were the most frequently used method for data collection in the included studies (n = 9) [11,28,31,33,35,37,39,40], followed by online questionnaires (n = 6) [14,29,32,34,36,38]. Two studies did not declare the method of data collection [30,41].

### 3.4. Moral Distress Levels

Table 3 shows the sample’s moral distress scores and characteristics in each included study.

Most of the studies included in this systematic review showed moderate moral distress among nurses [11,14,31,32,34,37,40]. While Asayesh et al. [41] highlighted the fact that nurses had a moderate-to-high level of moral distress [41], Browning and Cruz’s research [33], at the baseline, showed low levels of moral distress. In general, the highest moral distress composite scores were related to cost-control, end-of-life care [27,34,39], and futile care [34,37]. According to Ganz et al. [39], moral distress intensity was significantly correlated with frequency scores. 

The levels of moral distress frequency ranged from low [34,37,38,39] to high [30] among ICU nurses. The lowest levels of moral distress frequency were associated with the following items: disregard of staff/abusive patients [14,39], discontinuing treatment when the patient could not pay (Ganz et al.) [39], and physician requests to not talk about the dying process to terminally ill patients. The highest levels of moral distress frequency were associated with the following item: performing medical tests that were not needed [11,14,30,33,38,39,41]. The levels of moral distress intensity among ICU nurses ranged from moderate [37,38,39] to high [30,34].

The lowest intensity scores were associated with items such as requesting organ donations from family members of patients who were expected to die [38,39], preparing demented, elderly, do-not-resuscitate (DNR) patients for percutaneous endoscopic gastrostomy (PEG) insertion [39], or giving medications intravenously during a code with no compressions or intubation [14,30,38]. The highest-scoring intensity items included conducting cardiopulmonary resuscitation (CPR) to prolong life [12,14,36,37,39], working with unqualified physicians [11,14,33,34,39], and giving “false hope” to patients or family members [33].

### 3.5. Personal Correlates of Moral Distress

The studies included showed an association between demographic- or work-related variables and levels of moral distress. In general, moral distress scores showed a relationship with the intention to leave a position of employment, gender, age, and years of work experience. 

#### 3.5.1. Moral Distress and Socio-Demographic Variables 

Some studies showed that gender [31], and educational level [31] were independent correlates of moral distress scores. Most of the included studies showed that female nurses had higher levels of moral distress than did males [31], while Dyo et al. [36] showed that males had higher scores for moral distress. Johnson-Coyle et al. [29] and Elpern et al. found no statistically significant differences [37]. 

Age correlates with moral distress frequency [31,39], and moral distress scores [30]. Johnson-Coyle et al. [29], and Elpern et al. [37] did not find any differences with age.

#### 3.5.2. Moral Distress and Job-Related Variables 

Types of units (respiratory ICU, post-anesthesia care unit, neuro-intensive care unit, or the cardiac care unit) [34,39], and country [34] showed a significant correlation with moral distress scores. Regarding the professional category, nurses had higher moral distress scores than did physicians or other health professionals, but this was not statistically significant [27,29]. Indeed, according to Dodek’s findings [27,35], higher moral distress scores in nurses were associated with lower levels of decision attitude and social support and with higher levels of psychological stressors and psychological strain. On the same topic, Lamiani et al. [11] did not identify any differences in the moral distress scores of physicians compared with nurses. 

The years of experience as a nurse and having a greater responsibility in a nursing role (rather than being a staff nurse) were positively correlated with moral distress [27,37,38,39]. Nurses with lower numbers of years of job experience had the highest moral distress scores [14,29]. However, Asayesh et al. [41] showed that nurses with more than four years of experience in ICUs experienced higher moral distress scores. These findings were not confirmed by Dyo et al. [36]. In this study, there was no significant difference in moral distress scores when comparing years of experience, age, or education. Additionally, Browning and Cruz found no correlation between years of nursing experience and moral distress scores [33].

Also, nurses who intended to leave the ICU experienced higher moral distress scores [14,27,36]. The findings of Browning and Cruz [33] did not confirm this correlation. Some studies showed a positive and significant correlation between professional autonomy [40], empowerment [32], and moral distress scores. Additionally, nurses who felt supported by head nurses experienced lower moral distress scores [30]. According to these findings, Papathanassoglou et al.’s study showed a negative association between moral distress scores and nurse–physician collaboration scores [34]. Additionally, this study showed significant associations between the frequency of morally distressing events, patient-to-nurse ratios, and perceived job status [34].

Another moral distress correlation was found with the ethical climate score [32]; where there was a positive ethical climate, moral distress levels were lower. Fumis et al. [28] and Johnson-Coyle [29] showed that moral distress is associated with severe burnout. According to Elpern et al., job satisfaction, retention, psychological and physical well-being, self-image, and spirituality are negatively affected by the moral distress experienced by nurses [37]. 

### 3.6. Quality Assessment 

Table 4 shows the quality assessments. Regarding the quality of the included studies in this systematic review, most of them scored as “good quality” [29,30,31,32,33,34,37,39,40], and four studies were of “satisfactory quality” [14,36,38,41].

Issues detected related to the selection parameter of the NOS scale, explicitly regarding (a) “sample size”, as some studies did not justify that; (b) “representativeness of the sample”, as some studies used a selected group of users or did not describe the sampling strategy; and (c) “non-respondents”, as the response rate of some studies was unsatisfactory; the comparability between respondents and non-respondents was unsatisfactory; or there was no description of the response rate or the characteristics of the responders and non-responders.

## 4. Discussion

Despite the rich literature on the topic, moral distress is still a relevant problem in clinical settings, increasing the risks of leaving the profession and burnout for healthcare professionals, as well as negative consequences for both the practitioner and the patient [9,45,46]. This is particularly true in ICUs due to the complexity of patients and a greater chance of the onset of ethical and moral distress conflicts [47]. Based on a previous systematic review [10], this systematic review aimed to summarize the literature regarding moral distress among nurses working in ICUs. A total of 17 studies were included. The findings of this systematic review showed a moderate level of moral distress among nurses working in ICUs. The results of this systematic review confirm that there are a lot of triggers of moral distress related to patient-level factors, unit/team-level, or system-level causes, according to previous studies [6,7,15,24].

Indeed, regarding patient-level factors, several studies included in this systematic review documented a high level of moral distress when the family requires aggressive therapy and treatments that induce excessive pain. 

Moral distress is often linked to unit/team-level and system-level causes. When there is poor communication in the unit or inadequate collaboration between team members, they could experience moral distress. Additionally, perceiving unsafe or incompetent staffing is documented as a cause of moral distress in several articles included in this systematic review Papathanassoglou et al. showed a negative association between moral distress scores and nurse–physician collaboration scores [34]. Additionally, this study showed significant associations between the frequency of morally distressing events, patient-to-nurse ratios, and perceived job status [34]. Finally, most studies showed that the highest moral distress scores were related to cost-control, end-of-life care, and futile care [27,29,34,37].

Beyond the triggers of moral distress, this systematic review showed some correlates of moral distress. The included studies showed that nurses working in the ICU with less work experience [14,27,36], who are younger [30,31,39], female [31], and who intend to leave [34] have higher levels of moral distress. This systematic review’s findings show a positive correlation between professional autonomy [40], empowerment [32], and moral distress scores. Additionally, nurses who feel supported by head nurses experienced lower moral distress scores [30]. 

This systematic review has limitations as we have collected all evidence only in English or Italian, representing a possible publication bias in summarizing the evidence. Therefore, other studies may have been conducted and published as grey literature; these are not included. Furthermore, the search was limited to only registered nurses and thus did not include advanced nurse practitioners or other healthcare workers. Another limitation is the analysis of the MD phenomenon by using only the Corley scale]. The authors chose this tool because it has been validated in many languages, and it is widely used in ICU settings. Another research gap is the under-representation of European countries. 

MD occurs mainly in ICUs, but it would be interesting to study the phenomenon in other settings to help understand whether the Corley scale is also suitable for different care settings and to analyze MD in other professional categories. Qualitative studies should be organized, allowing researchers to collect experiences directly from healthcare professionals. Further research should investigate the role of spirituality or religion on moral distress scores. 

This systematic review summarized the evidence regarding moral distress in ICUs. By conducting a meta-analysis, it may be interesting to explore which type of nurse characteristics are associated positively or negatively with moral distress scores. To this point, we tried to obtain some statistical analyses in order to summarize the relationship between each individual’s moral distress scores and her personal or professional characteristics and to evaluate the association between demographic- or work-related variables and levels of moral distress. However, it is necessary to observe that there is extreme heterogeneity between the studies. Even if we included only studies using the Moral Distress Scale based on Corley’s theory, most included studies did not show the mean score of frequency or intensity. This prevented us from conducting a meta-analysis and a forest plot, and we could not retrieve these data from the authors because they often did not answer our emails. Further research should investigate the role of spirituality or religion on moral distress scores. 

## 5. Conclusions

Suggestions for the prevention of moral distress have emerged from the analyzed studies, which could allow the early identification of those who are particularly at risk of moral distress [48] and the strategies to be implemented. Organizations and professional associations should pay attention to nurses with lower levels of job experience [14,29] and nurses with long-standing expertise and positions of responsibility [27,37,38,39]. Indeed, considering the significant amount of time a nurse needs to acquire the necessary skills to work independently in the intensive care area [49,50], the organization would be wasting time and economic resources [51,52,53]. A critical care nurse leaving a clinical setting may result in a considerable increase in costs for the organization and an increase in risks for patients. These are more likely to be cared for by inexperienced or inexperienced staff [35], who are at risk of developing moral distress early on [14,29]. 

The organization should monitor the ethical climate [32] and support inter-professional communication [34] to keep moral distress levels low. The positive correlation between professional autonomy, empowerment, and moral distress scores may lead managers, educators, and healthcare professionals to make decisions to ensure a training or an organizational system that improves autonomy by reducing the risk of developing moral distress.

Unfortunately, organizations are facing a significant challenge. Since the beginning of the COVID-19 pandemic, several studies have highlighted the potential for moral distress among healthcare workers due to resource constraints, policies that promote safety but limit patient contact and social relationships, and the need to protect against the risk of infection [8,49,54,55]. Therefore, the results of this review are even more critical to orienting organizational strategies in the coming years.

## Figures and Tables

**Table 1 ijerph-19-10640-t001:** Inclusion and exclusion criteria.

PEOS Strategy	Inclusion Criteria	Exclusion Criteria
P—*Population*	Registered nurses	Physician; student; trainer; pharmacist; Any healthcare professional who is not a nurse.
E—*Exposure*	Moral Distress	Intention to leave; burnout; work-related stress
O—*Outcome*	Any measure related to the Moral Distress Scales developed by Corley et al.	Studies that do not report any moral distress score as a primary outcome
S—*Setting*	Intensive care units for adults	Emergency department; medical wards; surgical wards; primary care; critical care Any ward for pediatric patients

**Table 2 ijerph-19-10640-t002:** Characteristics of included studies.

Author (Year of Publication)	Title	Country	Study Design	Survey
Moral Distress Measure: Moral Distress Scale–Revised with 21 items				
Dodek, et al. [27]	Moral distress in intensive care unit professionals is associated with profession, age, and years of experience	Canada	Descriptive cross-sectional	Paper questionnaire
Fumis, et al. [28]	Moral distress and its contribution to the development of burnout syndrome among critical care providers.	Canada	Descriptive cross-sectional	Paper questionnaire
Johnson-Coyle, et al. [29]	Moral distress and burnout among cardiovascular surgery intensive care unit healthcare professionals: A prospective cross-sectional survey	Canada	Descriptive cross-sectional	Online survey
Ameri, et al. [30]	Moral distress and the contributing factors among nurses in different work environments	Iran	Descriptive cross-sectional correlational	/
Soleimani, et al. [31]	Spiritual well-being and moral distress among Iranian nurses	Iran	Descriptive cross-sectional	Paper questionnaire
Altaker, et al. [32]	Relationships among palliative care, ethical climate, empowerment, and moral distress in intensive care unit nurses.	USA	Descriptive cross-sectional correlational	Online survey
Browning, et al. [33]	Reflective debriefing: A social work intervention addressing moral distress among ICU nurses.	USA	Quasi-experimental	Paper questionnaire
Papathanassoglou, et al. [34]	Professional autonomy, collaboration with physicians, and moral distress among European intensive care nurses.	Croatia, Cyprus, Netherlands, Slovenia, United Kingdom, Belgium	Descriptive cross-sectional	Online survey
Dodek, et al. [35]	Moral distress is associated with general workplace distress in intensive care unit personnel.	Canada	Descriptive cross-sectional	Paper questionnaire
Moral Distress Scale with 38 items				
Dyo, et al. [36]	Moral distress and intention to leave: A comparison of adult and pediatric nurses by hospital setting.	USA	Descriptive cross-sectional correlational	Online survey
Elpern, et al. [37]	Moral distress of staff nurses in a medical intensive care unit.	USA	Descriptive cross-sectional	Paper questionnaire
Sauerland, et al. [38]	Assessing and addressing moral distress and ethical climate, part 1.	USA	Descriptive cross-sectional correlational	Online survey, by institutional email
Ganz, et al. [39]	Moral distress and structural empowerment among a national sample of Israeli intensive care nurses.	Israel	Descriptive cross-sectional correlational	Paper questionnaire
Moral Distress scale–Revised with 18 items				
Lusignani, et al. [14]	Moral distress among nurses in medical, surgical, and intensive-care units	Italy	Descriptive cross-sectional	Online survey
Yeganeh, et al. [40]	The relationship between professional autonomy and moral distress in ICU nurses of Guilan University of Medical Sciences in 2017	Iran	Descriptive cross-sectional correlational	Paper questionnaire
Moral Distress Scale–Revised with 14 items				
Lamiani, et al. [11]	Clinicians’ moral distress and family satisfaction in the intensive care unit.	Italy	Descriptive cross-sectional	Paper questionnaire
Moral Distress Scale–Revised with 30 items				
Asayesh, et al. [41]	The relationship between futile care perception and moral distress among intensive care unit nurses.	Iran	Descriptive cross-sectional correlational	/

**Table 3 ijerph-19-10640-t003:** Moral distress scores and sample characteristics.

Author (Year of Publication)	Mean Score Scale (SD), Range	Mean of Frequency	Mean of Intensity	Sample	Female/Male	Mean Age (SD), Range	Mean Years of Job Experience as Nurse (SD), Range
Elpern, et al. [37]	/	mean: 1.73 (range 0.74–4.42; SD 0.90)	mean: 3.66 (range 1.76–5.79; SD 1.73)	28	28/6	Age, years: 20–30: 40% of responders; 31–40: 24% of responders; 41–50: 28% of responders; 51–60: 8% of responders;	mean: 9.24
Papathanassoglou, et al. [34]	mean, 73.67; SD, 39.19; scale range, 0-336	mean 25.46; SD, 11.89; scale range, 0–84	mean, 56.99; SD, 16.76; scale range, 0–84	255	212/43	/	/
Ganz, et al. [39]	/	1.5 ± 0.7	3.7 ± 1.4	291	210/69 (12 missing data)	37.9 ± 8.9	13.9 ± 9.6
Sauerland, et al. [38]	/	2.86 (SD, 1.88) to 0.23 (SD, 0.93)	3.79 (SD, 2.21) to 2.14 (SD, 2.42)	225	181/44	ranged in age from 21 y to more than 60 y, with the majority (n = 144; 64%) between the ages of 30 to 49 y	15.67 ± 10.09)
Ameri, et al. [30]	mean = 3.29 (SD: 1.49)	Low: 2 (5.26); Moderate: 16 (42.10)); High: 20 (52.63)	9.93 (1.62)	38	/	/	/
Dodek, et al. [35]	83 (55, 119) 76 (48, 115) 57 (45, 70)	/	/	428	372/56	41(10)	5 (2, 11) Median
Dyo, et al. [36]	/	1.6 (0.11); % Nurses score ≥ 2.0 Mod-high freq = 24.6%	2.5 (0.19); % Nurses score ≥ 4.0 = 21.3%	279	259/20	43.6 years (12.0) [23–69 years]	/
Johnson-Coyle, et al. [29]	Nurse Median IQR: 80 (57–110) range: 5–246	/	/	129	109/21	26 had < 25 years; 58 had from 26 to 34 years; 34 had from 35 to 50 years; 11 had > 51 years	27 had <2 y; 36 had from 2 to 5 y; 21 had from 6–10 y; 45 had more than 10 y.
Fumis, et al. [28]	107.7 (60.4)	40.0 (13.7)	40.7 (17.8)	/	/	/	/
Lusignani, et al. [14]	/	/	/	/	/	/	16 (1–38) y
Altaker, et al. [32]	mean = 96.5 (SD: 55.8), 0–225	/	/	238	214/24	38 (11), 20-70	12 (11), <1–49
Asayesh, et al. [41]	137.53 (23.14).	/	/	117	78/39	34.99 (7.34)	/
Browning, et al. [33]	M = 81.81, SD = 37.43	/	/	30 control group; 6 experimental group	/	/	9 had 0–5 y, 11 had 6–10 y, 2 had 11–20 y, and 6 reported having over 20 y.
Dodek, et al. [35]	83 (55, 119)	/	/	428	372/56	41(10)	5 (2, 11) Median
Soleimani, et al. [31]	Total moral distress (mean—SD): 118.86 (69.62)	/	/	58	/	/	/
Yeganeh, et al. [40]	Total moral distress (mean—SD): 140.85 ± 5.45	48.18 ± 13.02	46.25 ± 10.82	180	169/11	34.75 ± 5.79	10.85 ± 5.09
Lamiani, et al. [11]	3.4 (SD = 1.5; median = 3.2; IQR=7.7).	/	/	77/24 drop	41/36	39.6 (7.38)	/

**Table 4 ijerph-19-10640-t004:** NOS for the risk of bias and quality assessment. The explanation of * and ** is on Appendix A.

	Selection				Comparability	Outcome		Total Score
Author	Representativeness of the Sample	Sample Size	Non-Respondents	Ascertainment of the Exposure (Risk Factor)	Confounding Factors Controlled	Assessment of Outcome	Statistical Test	
Elpern, et al. [37]		*	*	**	*	*	*	7
Papathanassoglou, et al. [34]		*	*	**	*	*	*	7
Ganz, et al. [39]	*	*		**	*	*	*	7
Sauerland, et al. [38]	*			**	*	*		5
Ameri, et al. [30]	*		*	**	*	*	*	7
Dodek, et al. [27]	*	*	*	**	*	*	*	8
Dyo, et al. [36]	*			**	*	*	*	6
Johnson-Coyle, et al. [29]	*	*	*	**	*	*	*	8
Fumis, et al. [28]	*		*	**	*	*	*	7
Lusignani, et al. [14]	*			**	*	*	*	6
Altaker, et al. [32]	*	*	*	**	*	*	*	8
Asayesh, et al. [41]	*		*	**	*	*		6
Browning, et al. [33]	*	*	*	**	*	*	*	8
Dodek, et al. [35]	*	*	*	**	*	*	*	8
Soleimani, et al. [31]	*	*	*	**	*	*	*	8
Yeganeh, et al. [40]	*	*	*	**	*	*	*	8
Lamiani, et al. [11]	*		*	**	*	*	*	7

## Data Availability

Not applicable.

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
