# Peer review of "Moral Distress Scores of Nurses Working in Intensive Care Units for Adults Using Corley’s Scale: A Systematic Review"

_ijerph, 2022, doi:10.3390/ijerph191710640_

Round 1

Reviewer 1 Report

Thank you for the opportunity to review this article. The time involved in submitting your manuscript is greatly appreciated.

Despite this, the article presents a series of issues that must be noted and mended. The recommendations are presented separately in sections. Hopefully, they would be useful.

Title: the title does not adequately reflect the content of the paper. Please, try to change it to better inform the readers about the type of studies that you included in the review (only those that measure Moral distress with a specific scale).

Abstract:

Less information appears in the abstract. Maybe expanded by adding the most relevant findings. Please, take into account that the abstract is the unique part of your paper that most of the readers could read. Hence, more information would be better. 

Introduction:

Firstly, some of the references that you cite are old. Even though the most relevant studies should be referenced, also the RECENT research must be included. Moreover, moral distress would be intensified during the COVID-19 Pandemic, but there is absent any mention of this fact in your paper.

Second, in lunes 86-88, you said that “The instruments developed by Corley and colleagues [5,23] are the most popular 86 instruments used in the literature”, but finally you identified only 17 articles using this instrument from 1863. It seems that the instruments developed by Corley are not extensively used. Please, could you better justify your decision to include ONLY Corley’s instruments?

At the end of the literature review, the aims and the questions in the research should appear. Maybe formulating the questions as a hypothesis would be an option to clear this aspect. Another commentary is the possibility of including this part in the final of the introduction part; even a separate section could be a good option, in order to clear the final of the introduction and to serve as a connection with the method.

Results

A lot of information is missing in 3.1. section. The reader can understand that you excluded 1863 papers, but he/she also wants to know why were excluded. If one of the characteristics of systematic reviews is replicability, you should give the readers more information on the process.

Moreover, are you sure that only Reading the Title and the abstract is enough for taking the decision of exclusion? Please, justify this point for readers.

In the same sense, about the 21 articles that don’t fit the inclusion criteria, could you give us more information?

The results should be presented in the same order as the introduction and hypotheses. Also, the same order must be used in the Tables. This simplifies the work for readers.

Finally, the repetition is constant all over the article. Please, try to change the words in order to do the reading more interesting and motivating

Discussion:

First of all, try to better adjust your conclusions to the findings. Or to say in other words, please try to justify more clearly the connection between your conclusions and your findings.

Finally, the section related to limitations, future lines of investigations, and the principal contributions of the research could be expanded. Your paper has a lot of relevant implications for society and policymakers, but you need to elaborate more on this topic.

Conclusion:

They don’t appear to new conclusions on this part. This part does not add any new to the rest of the paper. Please, try to condense your findings, or highlight your main contribution to the field.

Minor points

In line 24, there is a number “2” in the abstract that seems an error. 

Author Response

Thank you for your precious suggestions. We done our best in order to improve the quality of the paper.

Attached the responses.

Kind regards

Reviewer 2 Report

Dear Authors,

I read your manuscript entitled '“Moral distress scores of nurses working in intensive care units for adults: A systematic review". The topic you choose is very important not only from the theoretical perspective but mostly from the practical point of view. Unfortunately, much research shows that high level of distress force nurses to leave their profession. The hospital managers, yet the goverment offices for the health issues in many countries, still did not offer any health promotion programm dedicated to the health care professionals. Your manuscript, the introduction, and the methodological part have great value and quality. You in detail described the methodology, searching strategy, and inclusion criteria. In the discussion, you summarized what we know (from the revised articles) about the problem, and what we can do. I can only wish you all the best in the writing process. I would recommend your article for further proceeding.

Author Response

(The authors gave the same response as above.)

Reviewer 3 Report

Thanks for the opportunity to review this engaging and well-written paper.

The theme is very relevant to nursing education and practice. The study is theoretically well grounded, and the methodology is adequately described. The aim of the study is well established, and the results and the discussion allow to answer the main goals and reflect the main findings. Yet, I suggest the authors revise the references accordingly to the norms (e.g., page 6, lines 237, 241).

Very important: Authors must revise the abstract, as it seems inadequate for this study. The abstract must be very clear and adequately written so that the readers can understand the Background, the primary goals, the methodology used, the main findings, the study's novelty, the conclusion, and the implications for theory and practice.

Author Response

(The authors gave the same response as above.)

Reviewer 4 Report

I have completed my review of the manuscript titled “Moral distress scores of nurses working in intensive care units 2 for adults: A systematic review”.

This topic is a very interesting and very important segment for the nurses working in intensive care units. Despite that, this study has some serious limitations.

The abstract is very insufficient. There were not any results. The abstract contains only the method of literature research. This is absolutely unacceptable.

The authors write: “This systematic review aimed to summarize the correlates of moral distress such as age, intention to leave a position of employment, gender, years of work experience, type of ICU, professional autonomy, empowerment, patient to nurse ratios, perceived job status, ethical climate score, burnout, job satisfaction, psychological and physical well-being, self-image, and spirituality

This is a very, very unspecified aim, is not clear, a not include a research question. Also, the research questions should be based on the PICO formulation. There is no PICO methodology.

The picture of the flowchart PRISMA search procedure and eligibility reporting is missing!! Forest plot of effect sizes and overall summary effect from studies, were also missed.

Information on participants across studies, main results, and the summary was also missed! Existing tables are very confusing and difficult to read. There are no summary results and summary conclusions.

Systematic review needs to present as the best form of evidence because it is positioned at the top of the hierarchy of evidence. This article has completely insufficient methodology and results. Despite a very important topic, and a very interesting idea this article had very serious limitations. According to all the already mentioned crucial facts, this paper does not seem suitable for publication in this journal.

Author Response

(The authors gave the same response as above.)

Round 2

Reviewer 4 Report

I have completed my repeat review of the manuscript titled “Moral distress scores of nurses working in intensive care units for adults using the Corley’s scale: A systematic review”.

The last version is significantly improved. But, still missing Forest plot of effect sizes and overall summary effect from studies. The authors` response to this comment, does not give the answer to this.

It is not entirely clear, based on which statistical methods the authors got the conclusion they state in summary and at the beginning of the discussion.

Although the authors provide from line 192 to line 199 a description of the statistical methods which they used, this is not visible in any table, while the summarized result of all studies presented, is missed. It seems, that those conclusions were based only on their interpretation, clearly, based on the results in the studies found. According to that, please, give the overall summary result.

Author Response

Dear reviewer

we tdone our best in order to improve the quality of the study and respond to your answer.

Thank you for your time
